# Remdesivir Treatment Lacks the Effect on Mortality Reduction in Hospitalized Adult COVID-19 Patients Who Required High-Flow Supplemental Oxygen or Invasive Mechanical Ventilation

**DOI:** 10.3390/medicina59061027

**Published:** 2023-05-26

**Authors:** Chienhsiu Huang, Tsung-Lung Lu, Lichen Lin

**Affiliations:** 1Department of Internal Medicine, Dalin Tzu Chi Hospital, Buddhist Tzu Chi Medical Foundation, Chiayi 622, Taiwan; 2Department of Nursing, Dalin Tzu Chi Hospital, Chiayi 622, Taiwan; dm007424@tzuchi.com.tw (T.-L.L.); df462594@tzuchi.com.tw (L.L.)

**Keywords:** COVID-19, remdesivir, hospital mortality, ordinal scale, oxygen requirement

## Abstract

*Background and Objectives*: The therapeutic impact of remdesivir on hospitalized adult COVID-19 patients is unknown. The purpose of this meta-analysis was to compare the mortality outcomes of hospitalized adult COVID-19 patients receiving remdesivir therapy to those of patients receiving a placebo based on their oxygen requirements. *Materials and Methods*: The clinical status of the patients was assessed at the start of treatment using an ordinal scale. Studies comparing the mortality rate of hospitalized adults with COVID-19 treated with remdesivir vs. those treated with a placebo were included. *Results*: Nine studies were included and showed that the risk of mortality was reduced by 17% in patients treated with remdesivir. Hospitalized adult COVID-19 patients who did not require supplemental oxygen or who required low-flow oxygen and were treated with remdesivir had a lower mortality risk. In contrast, hospitalized adult patients who required high-flow supplemental oxygen or invasive mechanical ventilation did not have a therapeutic benefit in terms of mortality. *Conclusions*: The clinical benefit of mortality reduction in hospitalized adult COVID-19 patients treated with remdesivir was associated with no need for supplemental oxygen or requiring supplemental low-flow oxygen at the start of treatment, especially in those requiring supplemental low-flow oxygen.

## 1. Introduction

Coronavirus disease 2019 (COVID-19) presents problems for healthcare systems, economies, and various societies. Patients infected with the severe acute respiratory syndrome coronavirus type 2 (SARS-CoV-2) may not present any symptoms at all or they may develop severe illness and require mechanical ventilation. The COVID-19 vaccination has been administered to at least 69.7% of people worldwide. In low-income nations, 27.8% of people have received at least one dose; healthcare resources are scarce, and many people have not received vaccinations [1]. Thus, antiviral therapy for COVID-19 infection continues to be a crucial component of disease management. Remdesivir transforms into an adenosine triphosphate analog and inhibits the RNA-dependent RNA polymerase (RdRp) of the virus by interfering with viral replication. Remdesivir has demonstrated antiviral activity against SARS-CoV-2, as well as against a wide variety of RNA virus families [2,3,4,5].

Remdesivir received early approval as a COVID-19 infection therapy by the U.S. Food and Drug Administration (FDA) and the European Medicines Agency (EMA) [6,7]. The FDA approved the use of remdesivir after reviewing three randomized controlled trials (RCTs) involving patients hospitalized with mild-to-severe COVID-19 infection. The Adaptive COVID-19 Treatment Trial (ACTT-1) showed that the median time to recovery from COVID-19 for the remdesivir group was 10 days as opposed to 15 days for the placebo group. The study also revealed that patients receiving remdesivir had a 28-day mortality rate of 11.6%, whereas those receiving placebo had a mortality rate of 15.5%. This impact was particularly seen in individuals who were requiring supplemental low-flow oxygen at the start of treatment. No mortality benefit was observed for those in whom remdesivir was initiated after the start of invasive mechanical ventilation [8]. In a second RCT, individuals with moderate COVID-19 infection who were hospitalized and received remdesivir treatment for five or ten days were compared those who received standard of care. At Day 11, the COVID-19 symptoms in the five-day treatment group improved significantly more than those in the standard-of-care group, but there was no clinically significant difference [9]. Five- and ten-day remdesivir treatment were contrasted in a third RCT of severe hospitalized COVID-19 adult patients. At the start of treatment, researchers assessed the clinical condition of the participants. The recovery or mortality rates between the two groups did not statistically significantly differ [10]. Regarding mortality, the FDA’s approval of remdesivir was based on the findings of three RCTs, and only the ACTT-1 demonstrated the benefit of reduced mortality risk in patients receiving supplemental low-flow oxygen at the start of treatment.

The EMA has also approved remdesivir as a treatment for COVID-19 infection. It is administered to adults and children who require supplemental low- or high-flow oxygen or other noninvasive ventilation at the start of treatment. The EMA approved remdesivir based on the findings of three RCTs. The first study was the ACTT-1 [8]. In a second study, 584 outpatients with a high risk of hospitalization due to underlying health conditions were evaluated for the effectiveness of remdesivir. When administered for three days within seven days of symptom onset, remdesivir reduced the risk of hospitalization by 87%. Remdesivir-treated patients were hospitalized at a rate of 0.7% over the course of 28 days, as opposed to 5.3% of placebo-treated patients [11]. A third study revealed that the use of remdesivir was well accepted and did not raise any new safety concerns in children [12]. Regarding mortality, the EMA approved remdesivir based on the same evidence as the FDA.

The publication of the final concluding findings of the World Health Organization (WHO) Solidarity trial showed that, overall, 602 of the 4146 patients assigned to the remdesivir group died compared to 643 of the 4129 patients assigned to the control group (*p* = 0.12). Among mechanically ventilated patients, 151 of the 359 assigned to the remdesivir group died compared to 134 of the 347 assigned to the control group (*p* = 0.32). Among those who were not on mechanical ventilation but required supplemental oxygen, 426 of the 2918 patients assigned to the remdesivir group died compared to 476 of the 2921 patients assigned to the control group (*p* = 0.03). Among those who did not initially require supplemental oxygen, 25 of the 869 patients assigned to remdesivir group died compared to 33 of the 861 patients assigned to the placebo group (*p* = 0.30) [13]. The WHO Solidarity trial suggested that only nonventilated COVID-19 patients who need oxygen supplementation benefit from remdesivir, including those who require supplemental conventional oxygen and high-flow oxygen or noninvasive ventilation. The WHO initially recommended opposition to using remdesivir but now weakly recommends it depending on disease severity. The ACTT-1 and WHO Solidarity trial findings were inconclusive. There is uncertainty regarding the efficacy of remdesivir [8,13,14]. Is remdesivir effective in lowering COVID-19 patient mortality? We conducted this meta-analysis to evaluate mortality among patients treated with remdesivir based on oxygen requirements and aimed to assess the clinical mortality outcomes of hospitalized adult COVID-19 patients treated with remdesivir.

## 2. Method

### 2.1. Data Search and Extraction

All clinical studies published between 1 January 2020 and 28 February 2023 were located through a literature search of the PubMed, Web of Science, and Cochrane Library databases. We searched for “Remdesivir”, “Veklury”, “GS-5734”, “COVID-19”, “coronavirus 2019”, and “SARS-CoV-2.” RCTs that directly compared the clinical effectiveness of remdesivir to a placebo in the treatment of hospitalized adult COVID-19 patients were considered eligible for inclusion. Information on authors, region, median time of symptoms before first dose of remdesivir, mean age, ordinal scale of patients at the start of treatment, total number of remdesivir-treated patients, total number of placebo-treated patients, and hospital mortality or 28-day mortality was extracted. The clinical status of the patients was assessed at the start of treatment using a four-category ordinal scale as below: (1) not requiring supplemental oxygen; (2) requiring supplemental low-flow oxygen; (3) requiring non-invasive ventilation or high-flow oxygen; (4) requiring invasive mechanical ventilation or extracorporeal membrane oxygenation (ECMO).

### 2.2. Inclusion Criteria

The primary outcome was hospital mortality or 28-day mortality within subgroups stratified by oxygen requirements at the start of treatment. The studies were considered eligible for inclusion only if they directly compared the clinical effectiveness of remdesivir vs. a placebo in hospitalized adult COVID-19 patients. Studies that had any one or more of the following outcomes were included: hospital mortality or 28-day mortality, and ordinal scale of the patients at the start of treatment.

### 2.3. Exclusion Criteria

The effects on mortality reduction could not be evaluated separately for patients who required supplemental conventional oxygen and those who required high-flow oxygen or noninvasive ventilation in the WHO Solidarity Trial and we excluded the trial [14]. In addition, four studies were sub-studies of the WHO Solidarity Trial [15,16,17,18,19], including three studies that were continuous studies [15,16,17,19].

### 2.4. Quality Assessment and Statistical Analysis

We assessed the risk of bias for each trial using the Cochrane Risk of Bias Tool 2.0 for RCTs.

Statistical analysis was completed using RevMan 5, the Cochrane Review Manager tool. For continuous and categorical variables, the relative risk (RR) and mean difference with a 95% confidence interval (CI) were calculated, respectively. Significant heterogeneity between the studies was defined as an I^2^ greater than 50% and a *p* value for the Q-test less than 0.10 for each study. When effects were thought to be homogenous, the fixed-effects model was applied, and when they were heterogeneous, the random-effects model was applied.

## 3. Results

### 3.1. Characteristics of the Included Studies

From the PubMed, Web of Science, and Cochrane Library databases, there were 914, 998, and 367 initial search results, respectively. There were 62 potentially relevant articles left after removing duplicates and irrelevant studies. Fifty-three articles were omitted from a full-text review, including the absence of remdesivir vs. placebo results for hospitalized adult COVID-19 patients and the study of WHO Solidarity Trial. Nine studies were included in the final meta-analysis (Figure 1) [8,9,15,17,18,19,20,21,22]. Table 1 displays the primary characteristics of the nine included studies. Figure 2 shows the assessment of the bias risk.

### 3.2. Effectiveness Outcomes

Nine studies including 4381 patients (2230 who received remdesivir therapy, 2151 who received placebo therapy) reported hospital mortality or 28-day mortality rates [8,9,15,17,18,19,20,21,22]. The mortality rate was significantly different between remdesivir-treated and placebo-treated patients (RR = 0.83, *p* = 0.02, I^2^ = 0%) (Figure 3). Three studies including 656 patients with no supplemental oxygen at the start of treatment (339 who received remdesivir therapy, 317 who received placebo therapy) reported hospital mortality or 28-day mortality rates [8,9,17]. The mortality rate was not significantly different between remdesivir-treated and placebo-treated patients (RR = 0.66, *p* = 0.27, I^2^ = 0%) (Figure 4). However, all three studies supported the use of remdesivir, and there was a trend of reduced mortality risk among patients who did not require supplemental oxygen and received remdesivir therapy. Three studies including 1329 patients requiring supplemental low-flow oxygen at the start of treatment (695 who received remdesivir therapy, 634 who received placebo therapy) reported hospital or 28-day mortality rates [8,17,22]. The mortality rate was significantly different between remdesivir-treated and placebo-treated patients (RR = 0.59, *p* = 0.0009, I^2^ = 47%) (Figure 5). Three studies including 579 patients requiring supplemental high-flow oxygen or noninvasive mechanical ventilation at the start of treatment (295 who received remdesivir therapy, 284 who received placebo therapy) reported hospital or 28-day mortality rates [8,17,22]. The mortality rate was not significantly different between remdesivir-treated and placebo-treated patients (RR = 0.99, *p* = 0.96, I^2^ = 0%) (Figure 6). Two studies including 397 patients requiring invasive mechanical ventilation or ECMO at the start of treatment (189 who received remdesivir therapy, 208 who received placebo therapy) reported hospital or 28-day mortality rates [8,17]. The mortality rate was not significantly different between remdesivir-treated and placebo-treated patients (RR = 1.00, *p* = 0.98, I^2^ = 0%) (Figure 7). Five studies including 2489 patients not requiring supplemental oxygen or requiring supplemental low-flow oxygen at the start of treatment (1287 who received remdesivir therapy, 1202 who received placebo therapy) reported hospital or 28-day mortality rates [8,9,15,17,22]. The mortality rate was significantly different between remdesivir-treated and placebo-treated patients (RR = 0.64, *p* = 0.001, I^2^ = 13%) (Figure 8). Four studies including 1303 patients requiring high-flow supplemental oxygen, noninvasive mechanical ventilation, invasive mechanical ventilation, or ECMO at the start of treatment (644 who received remdesivir therapy, 659 who received placebo therapy) reported hospital or 28-day mortality rates [8,15,17,22]. The mortality rate was not significantly different between remdesivir-treated and placebo-treated patients (RR = 0.98, *p* = 0.88, I^2^ = 0%) (Figure 9).

## 4. Discussion

The current meta-analysis found that remdesivir treatment reduced the risk of hospital mortality compared to placebo treatment among hospitalized adult COVID-19 patients. Hospital mortality was at a 17% lower risk. Hospitalized adult COVID-19 patients who did not require supplemental oxygen or who required supplemental low-flow oxygen at the start of treatment and those treated with remdesivir had a lower risk of hospital mortality than those treated with a placebo. Hospital mortality was at a 36% lower risk. Hospitalized adult COVID-19 patients who required supplemental low-flow oxygen at the start of treatment and those treated with remdesivir had a lower risk of hospital mortality than those treated with a placebo. Hospital mortality was at a 41% lower risk. Therefore, the current meta-analysis found that hospitalized adult COVID-19 patients who required supplemental low-flow oxygen at the start of treatment had the best clinical benefit of reduced mortality risk.

Nine RCTs were included in the Cochrane Reviews performed by Felicitas Grundeis et al. (2023) and found that remdesivir was determined to make little or no difference in all-cause 28-day mortality [23].

### 4.1. Meta-Analysis Exploring Mortality in Hospitalized Adult COVID-19 Patients Receiving Remdesivir

The meta-analysis performed by Al-Abdouh et al. (2021) included four RCTs [8,9,14,22], revealing that patients who received remdesivir had higher rates of hospital discharge, but the risk of mortality did not significantly decrease [24]. The meta-analysis performed by Singh et al. (2021) included four RCTs [8,9,14,22], revealing that the remdesivir group showed no mortality benefit over the control group. There were higher rates of clinical improvement in the remdesivir group [25]. The meta-analysis performed by Lai et al. (2021) included five RCTs [8,9,10,14,22], revealing that remdesivir improves clinical outcomes in COVID-19 patients who are hospitalized. However, there was no mortality benefit observed with remdesivir therapy [26].

The meta-analysis performed by Gholamhoseini et al. (2022) included five RCTs and one observational cohort study [8,9,14,22,27,28], revealing that remdesivir had beneficial effects on clinical improvement but no appreciable impact on mortality after 14 days of treatment [29].

The meta-analysis performed by Tanni et al. (2022) included six RCTs and three observational cohort studies [8,9,14,15,21,22,30,31,32], revealing that in terms of mortality, the remdesivir and control groups did not differ statistically from one another. Patients with COVID-19 infection received remdesivir treatment, however there was no difference in clinically relevant results [33]. The meta-analysis performed by Kaka et al. (2022) included five RCTs [8,9,14,20,22], revealing that remdesivir therapy had little to no effect on mortality in hospitalized adult COVID-19 patients [34].

All of the above six meta-analyses showed that, compared to recipients of a placebo, hospitalized adult COVID-19 patients who received remdesivir had no reduction in mortality risk. In addition, all six meta-analyses included the WHO Solidarity Trial (weighted more than 60%) in the meta-analysis. There were 2743 patients who received remdesivir therapy and 2708 patients who received a placebo in the WHO Solidarity Trial, which showed that there was no significant impact on outcomes that are crucial for the patient, like mortality or the necessity for mechanical ventilation [14]. Therefore, hospitalized patients with COVID-19 who received remdesivir therapy showed no reduction in mortality risk in any of the six meta-analyses. The conclusions should be consistent with that of the WHO Solidarity Trial. Because of the large sample size of the WHO Solidarity Trial, this limitation may contribute to the same results in meta-analyses including the WHO Solidarity Trial.

The meta-analysis performed by Shrestha et al. (2021) included three RCTs [8,9,22], revealing that patients who received remdesivir exhibited decreased 14-day mortality rates, and enhanced clinical recovery. Clinical recovery was reported to have occurred earlier in patients treated with remdesivir [35]. The meta-analysis performed by Sarfraz et al. (2021) included four RCTs [8,9,22,29], revealing that remdesivir increases clinical benefits in COVID-19 patients by reducing mortality risk and the need for oxygen support [36]. These two meta-analyses excluded the WHO Solidarity Trial. Therefore, they showed that hospitalized adult COVID-19 patients who received remdesivir experienced a therapeutic advantage of lowered mortality risk.

### 4.2. Real-World Studies Exploring Mortality in Hospitalized Adult COVID-19 Patients Receiving Remdesivir

Olender et al. (2021) reported that in comparison to the non-remdesivir cohort, in the remdesivir cohort, the 28-day mortality rate was significantly lower (12.0% vs. 16.2%; *p* = 0.03) [32]. Chokkalingam et al. (2022) reported that there were 3557 mortality events (14.3%) in the remdesivir group, while there were 3775 in the control group (15.2%). Remdesivir treatment was connected to a statistically significant reduction of 17% in inpatient mortality in hospitalized adult COVID-19 patients when compared to control patients [37]. The study of Ohi et al. (2021) included 2374 patients who received remdesivir treatment and 3524 who did not receive remdesivir treatment. Remdesivir treatment was not linked to increased survival in this group of hospitalized US veterans COVID-19 patients [31].

We found conflicting results in real-world clinical studies related to hospitalized adult COVID-19 patients who received remdesivir.

According to the current meta-analysis, remdesivir therapy for hospitalized adult COVID-19 patients merely led to a 17% reduction in the likelihood of hospital mortality. The treatment outcomes were disappointing regarding the survival of hospitalized adult COVID-19 patients. When the current meta-analysis included the WHO Solidarity Trial (weight = 74.3%) and excluded the four sub-studies [15,17,18,19], six studies including 7412 patients (3769 who received remdesivir therapy, 3643 who received placebo therapy) reported hospital or 28-day mortality rates. The mortality rate was not significantly different between patients treated with remdesivir and placebo (RR = 0.94, 95% CI = 0.83–1.08, *p* = 0.39, I^2^ = 0%). Not all hospitalized COVID-19 patients treated with remdesivir have a clinical benefit of reduced mortality risk, but there must be a subgroup of patients with this clinical benefit. Identifying these subgroups was the purpose of this meta-analysis.

### 4.3. Oxygen Requirements of Hospitalized Adult COVID-19 Patients at the Start of Treatment

The meta-analysis performed by Beckerman et al. (2022) included six RCTs [8,9,14,21,22,38], revealing that remdesivir lowered the 28-day mortality rate among patients with low-flow oxygen. No improvement was observed among high-flow oxygen patients, including non-invasive mechanical ventilation [39]. In 2022, Lee et al. performed a meta-analysis including eight RCTs [7,8,13,14,17,20,21,22], and the results showed that remdesivir led to mortality RRs of 0.77 (95% CI, 0.5–1.19) in patients who did not require supplemental oxygen, 0.89 (95% CI, 0.79–0.99) in patients who were not ventilated but required oxygen, and 1.08 (95% CI, 0.88–1.31) in patients who were on mechanical ventilation. Remdesivir probably lowers mortality in COVID-19 patients who require supplemental oxygen but are not ventilated [40]. The meta-analysis performed by Amstutz et al. in 2023 included nine RCTs [8,9,13,14,15,17,18,19,22], and as opposed to 706 of the 5005 patients allocated to the non-remdesivir group, 662 of the 5317 patients assigned to the remdesivir group passed away (*p* = 0.045). Remdesivir provides significant survival benefits to hospitalized COVID-19 patients who did not require supplemental oxygen or who required conventional oxygen support [41]. Mozaffari et al. matched a total of 16,687 non-remdesivir patients and 28,855 remdesivir patients. In comparison to 19.1% of the non-remdesivir patients, 15.4% of the remdesivir patients passed away within 28 days. Overall, remdesivir was associated with a reduction in 28-day mortality. This 28-day mortality benefit was also observed for patients not requiring supplemental oxygen, requiring supplemental low-flow oxygen, and requiring invasive mechanical ventilation or ECMO. A lower risk of 28-day mortality was not observed in the supplemental high-flow oxygen or noninvasive mechanical ventilation remdesivir groups [42]. According to Garibaldi et al., there was no significant impact on the overall mortality rate in hospitalized adult COVID-19 patients. Patients requiring low-flow oxygen who received remdesivir were significantly less likely to die than controls. The routine initiation of remdesivir in patients already requiring supplemental high-flow oxygen, noninvasive mechanical ventilation or invasive mechanical ventilation is unlikely to be beneficial [43]. Tsuzuki et al. evaluated the effectiveness of remdesivir in hospitalized COVID-19 patients who did not require supplemental oxygen. According to the study, the 30-day mortality risk between the remdesivir group and the standard-of-care group did not differ significantly [44].

Based on literature findings and the current meta-analysis, it has been shown that hospitalized adult COVID-19 patients requiring supplemental low-flow oxygen at the start of treatment who receive remdesivir therapy experience a reduced mortality risk. There was no clinical benefit of reduced mortality risk in patients requiring supplemental high-flow oxygen, noninvasive mechanical ventilation, invasive mechanical ventilation, or ECMO in hospitalized adult COVID-19 patients who received remdesivir therapy. There may be a tendency toward a decreased risk of mortality among hospitalized adult COVID-19 patients who did not require supplemental oxygen and were treated with remdesivir.

### 4.4. Early Administration of Remdesivir for Adult COVID-19 Patients

According to Russo et al. (2022), remdesivir was a treatment option for 16,999 patients who were hospitalized for COVID-19 and required supplemental low-flow oxygen therapy. By Day 29, the mortality rate for patients with an interval of 0 to 2 days between hospital admission and drug prescription was 10.8%, while the mortality rate for patients with an interval of 3 to 5 days was 14.5%. According to the study of patients who received remdesivir, the length of time between a SARS-CoV-2 diagnosis and medication prescription significantly affects the mortality of hospitalized SARS-CoV-2 patients [45]. Falcone et al. (2022) reported that remdesivir was administered to 312 individuals with COVID-19 pneumonia; 90 (20.8%) received it early (before 5 days of onset of symptoms), and 222 (71.2%) received it late. In the early-remdesivir group, 29 patients (32.2%) progressed to severe COVID-19, compared to 104 patients (46.8%) in the late-remdesivir group (*p* = 0.018). The only protective factor was the early administration of remdesivir (OR = 0.49; *p* = 0.015). The author concluded that remdesivir should be used within five days of symptom onset to prevent the progression of COVID-19 [46].

The median interval between symptoms and the first dosage of remdesivir was between 5 and 10 days in the current meta-analysis. Most hospitalized adult COVID-19 patients received their first dose of remdesivir 5 days after symptom onset. The disappointing result of hospital mortality was at a 17% lower risk in the current meta-analysis. The delay in the first dose of remdesivir therapy should be an attribution factor. If these hospitalized adult COVID-19 patients received early remdesivir treatment, COVID-19 progression may have been reduced and survival may have increased.

### 4.5. Limitations

First, two of the studies included in the current meta-analysis had a high risk of bias. The population sizes and number of included studies were modest. The nine studies included in the current meta-analysis were heterogeneous due to different counties, populations, and study designs. Second, the current meta-analysis did not include data on vaccine history related to hospitalized adult COVID-19 patients. Therefore, we were unable to explore the interaction between remdesivir treatment and COVID-19 vaccination. Third, only five RCTs reported detailed patient data at the start of treatment using an ordinal scale. Therefore, we could not make definite conclusions regarding patients who did not requiring supplemental oxygen. To confirm the mortality benefit of remdesivir therapy in hospitalized adults with COVID-19 who do not require supplemental oxygen, additional RCTs are required. Fourth, we suppose that the early use of remdesivir in hospitalized adult COVID-19 patients may reduce COVID-19 progression and increase survival outcomes. Further RCTs are needed to confirm the association between the early use of remdesivir and mortality outcomes.

## 5. Conclusions

A clinical benefit of mortality reduction was observed in hospitalized adult COVID-19 patients who requiring supplemental low-flow oxygen at the start of treatment. There was no clinical benefit of reduced mortality risk in hospitalized adult COVID-19 patients requiring supplemental high-flow oxygen, noninvasive mechanical ventilation, invasive mechanical ventilation, or ECMO who were treated with remdesivir therapy. The level of supplemental oxygen in hospitalized adult COVID-19 patients may be a beneficial indicator for remdesivir treatment decisions. Patients with COVID-19 who require hospitalization may benefit clinically from early identification and early administration of remdesivir.

## Figures and Tables

**Figure 1 medicina-59-01027-f001:**
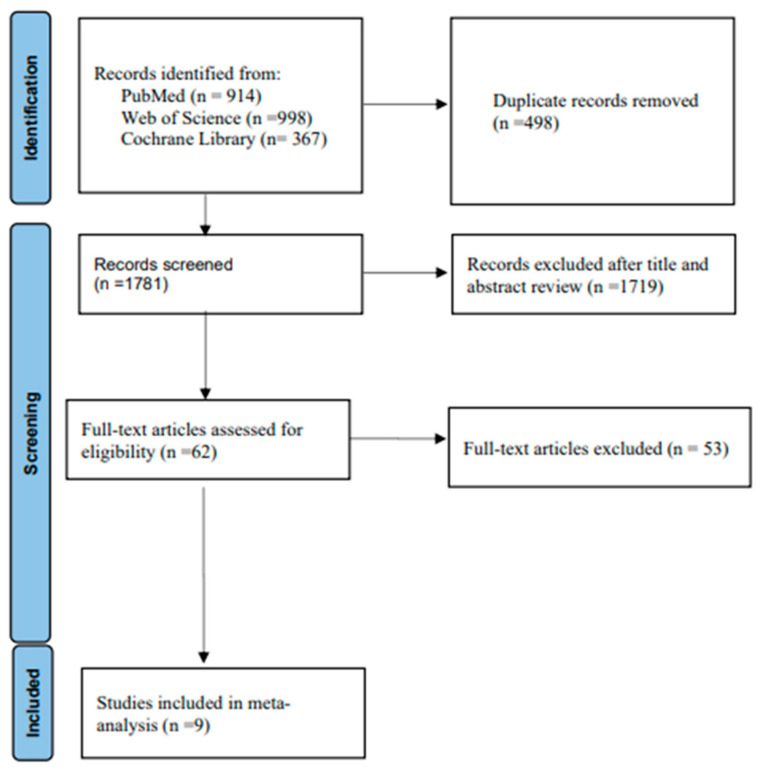
Flow diagram of the study selection process.

**Figure 2 medicina-59-01027-f002:**
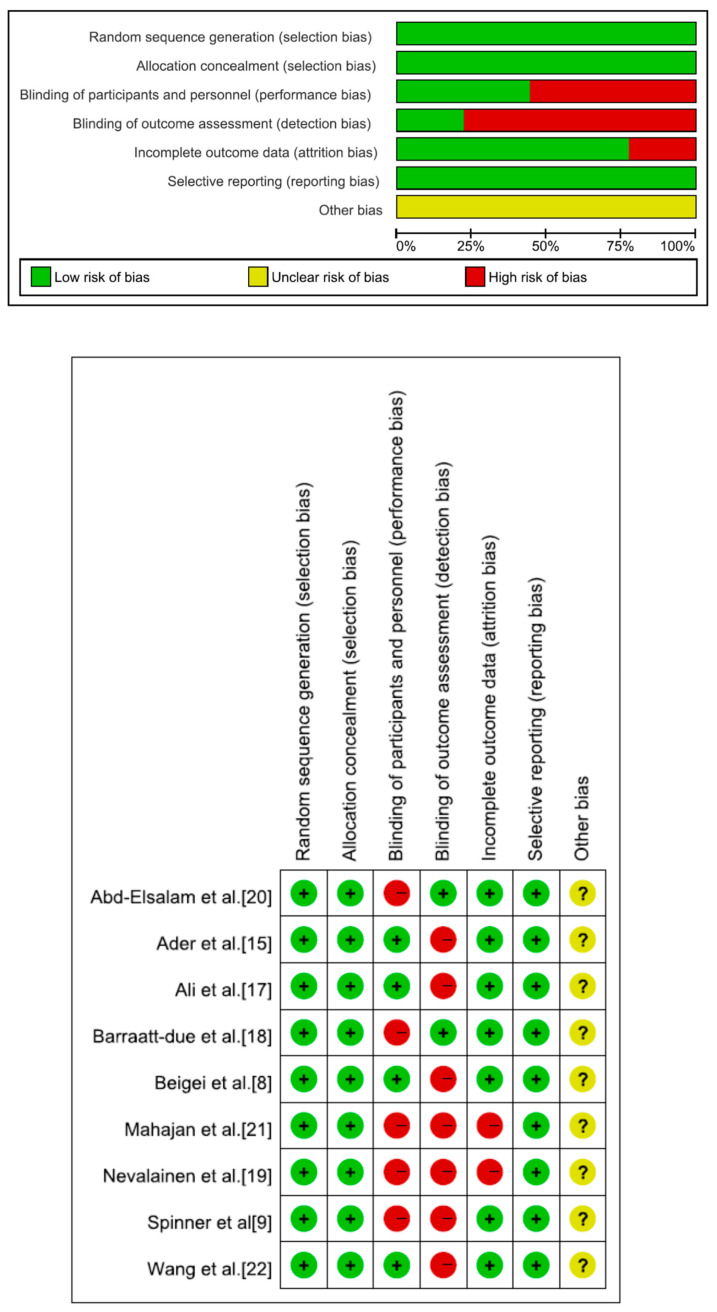
Risk of bias of nine randomized controlled trials and the studies of Mahajan et al. and Nevalainen et al with a high risk of bias.

**Figure 3 medicina-59-01027-f003:**
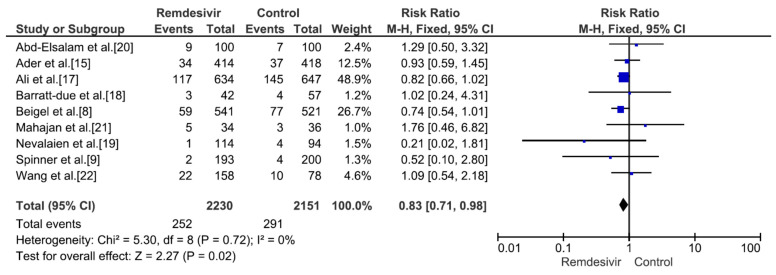
Hospital mortality or 28-day mortality between remdesivir and placebo in the treatment of hospitalized adult COVID-19 patients. Figure 3 legend: Nine studies including 4381 patients reported hospital mortality or 28-day mortality rates. The mortality rate was significantly different between remdesivir-treated and placebo-treated patients.

**Figure 4 medicina-59-01027-f004:**
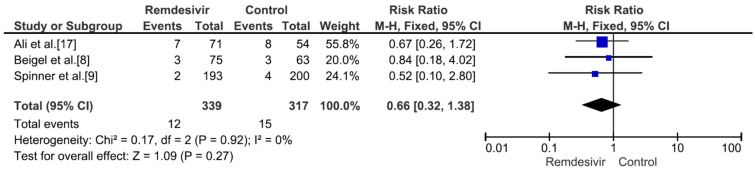
Hospital mortality or 28-day mortality between remdesivir and placebo in the treatment of hospitalized adult COVID-19 patients not requiring supplemental oxygen at the start of treatment. Figure 4 legend: Three studies including 656 patients with no supplemental oxygen at the start of treatment reported hospital mortality or 28-day mortality rates. The mortality rate was not significantly different between remdesivir-treated and placebo-treated patients.

**Figure 5 medicina-59-01027-f005:**
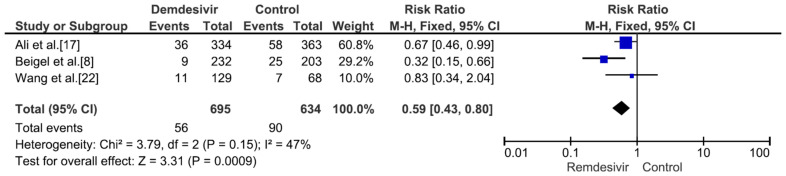
Hospital mortality or 28-day mortality between remdesivir and placebo in the treatment of hospitalized adult COVID-19 patients requiring supplemental low-flow oxygen at the start of treatment. Figure 5 legend: Three studies including 1329 patients requiring supplemental low-flow oxygen at the start of treatment reported hospital or 28-day mortality rates. The mortality rate was significantly different between remdesivir-treated and placebo-treated patients.

**Figure 6 medicina-59-01027-f006:**
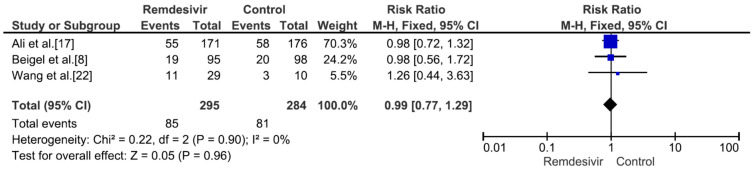
Hospital mortality or 28-day mortality between remdesivir and placebo in the treatment of hospitalized adult COVID-19 patients requiring supplemental high-flow oxygen or noninvasive mechanical ventilation at the start of treatment. Figure 6 legend: Three studies including 579 patients requiring supplemental high-flow oxygen or noninvasive mechanical ventilation at the start of treatment reported hospital or 28-day mortality rates. The mortality rate was not significantly different between remdesivir-treated and placebo-treated patients.

**Figure 7 medicina-59-01027-f007:**
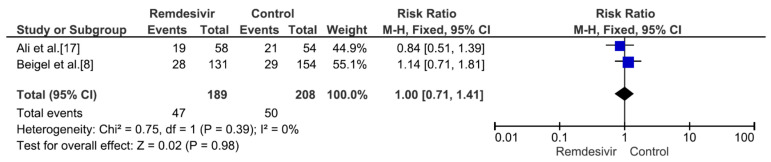
Hospital mortality or 28-day mortality between remdesivir and placebo in the treatment of hospitalized adult COVID-19 patients requiring invasive mechanical ventilation or extracorporeal membrane oxygenation at the start of treatment. Figure 7 legend: Two studies including 397 patients requiring invasive mechanical ventilation or ECMO at the start of treatment reported hospital or 28-day mortality rates. The mortality rate was not significantly different between remdesivir-treated and placebo-treated patients.

**Figure 8 medicina-59-01027-f008:**
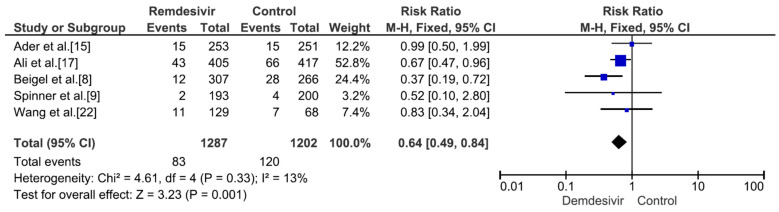
Hospital mortality or 28-day mortality between remdesivir and placebo in the treatment of hospitalized adult COVID-19 patients not requiring supplemental oxygen or requiring supplemental low-flow oxygen at the start of treatment. Figure 8 legend: Five studies including 2489 patients not requiring supplemental oxygen or requiring supplemental low-flow oxygen at the start of treatment reported hospital or 28-day mortality rates. The mortality rate was significantly different between remdesivir-treated and placebo-treated patients.

**Figure 9 medicina-59-01027-f009:**
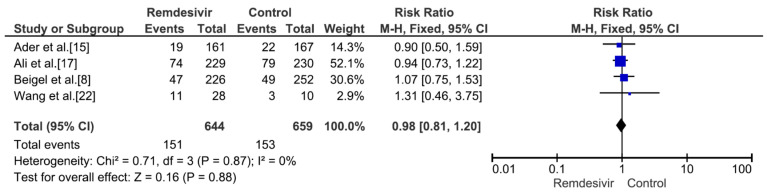
Hospital mortality or 28-day mortality between remdesivir and placebo in the treatment of hospitalized adult COVID-19 patients requiring high-flow supplemental oxygen, noninvasive mechanical ventilation, invasive mechanical ventilation, or extracorporeal membrane oxygenation at the start of treatment. Figure 9 legend: Four studies including 1303 patients requiring high-flow supplemental oxygen, non-invasive mechanical ventilation, invasive mechanical ventilation or ECMO at the start of treatment reported hospital or 28-day mortality rates. The mortality rate was not significantly different between remdesivir-treated and placebo-treated patients.

**Table 1 medicina-59-01027-t001:** Characteristics of the included studies.

Author Design	Region	Study Period	Number of Patients	Mean Age of Patients	Other Treatments for Patients Receiving Remdesivir	Median Time (IQR) of Symptoms before First Dose of Remdesivir
Abd-Elsalam et al. [20]	Egypt	June 2020 to Dec. 202	Rem: 100 Pla: 100	Rem:55.0 Y/O Pla:56.5 Y/O	No other treatment	3 days * (SD = 0)
Ader et al. [15]	Europe	Mar. 2020 to Jan. 2021	Rem: 414 Pla: 418	Rem:64.0 Y/O Pla:63.0 Y/O	No other treatment	9.0 days (7–11)
Ali et al. [17]	Canada	Aug. 2020 to April 2021	Rem: 634 Pla: 647	Rem:65.0 Y/O Pla:66.0 Y/O	Steroid (87.2%) Tocilizumab (2.2%)	8.0 days (5–11)
Barratt-due et al. [18]	Norway	Mar. 2020 to Oct. 2020	Rem: 42 Pla: 57	Rem:59.7 Y/O Pla:58.1 Y/O	Steroid (2.4%) Immunomodulatory drugs (2.4%)	7.5 days * (SD = 6.1)
Beigel et al. [8]	International	Feb 2020 to April 2020	Rem: 541 Pla: 521	Rem:58.6 Y/O Pla:59.2 Y/O	Steroid (21.6%) Hydroxychloroquine (34.6%) Immunomodulatory drugs (4.3%)	9.0 days (6–12)
Mahajan et al. [21]	India	June 2020 to Dec. 2020	Rem: 34 Pla: 36	Rem:58.1 Y/O Pla:57.4 Y/O	No report	6.8 days * (SD = 2.49)
Nevalainen et al. [19]	Finland	July 2020 to Jan. 2021	Rem: 114 Pla: 94	Rem:57.2 Y/O Pla:59.7 Y/O	No report	5.0 days (4–8)
Spinner et al. [9]	International	Mar 2020 to April 2020	Rem: 193 Pla: 200	Rem:56.0 Y/O Pla:57.0 Y/O	Steroid (15.0%) Hydroxychloroquine (34.6%) Tocilizumab (0.5%) Lopinavir-ritonavir (5.7%)	8.0 days (15–11)
Wang et al. [22]	China	Feb 2020 to Mar. 2020	Rem: 158 Pla: 78	Rem:66.0 Y/O Pla:64.0 Y/O	Steroid (64/6%) Lopinavir-ritonavir (27.8%)	10.0 days (9–12)

Foot notes: Rem: remdesivir; Pla: Placebo; Y/O: years old; IQR: interquartile range; *: mean days; SD: standard deviation.

## Data Availability

Data sharing not applicable.

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
