# Peer review of "Remdesivir Treatment Lacks the Effect on Mortality Reduction in Hospitalized Adult COVID-19 Patients Who Required High-Flow Supplemental Oxygen or Invasive Mechanical Ventilation"

_medicina, 2023, doi:10.3390/medicina59061027_

Round 1
Reviewer 1 Report
Dear authors, the article is well written, and I congratulate you for that. I understand the restriction on vaccination. I would still like you to tell us what other treatment these patients had, if it somehow influenced their evolution. I would like you to tell me if other associated comorbidities could influence the evolution of these patients, if there is any data on this. Adverse reactions were found?
Author Response
Reviewer1:
Dear authors, the article is well written, and I congratulate you for that. I understand the restriction on vaccination.
- I would still like you to tell us what other treatment these patients had, if it somehow influenced their evolution.
Reply: I add other treatments for patients receiving remdesivir in Table 1.
- I would like you to tell me if other associated comorbidities could influence the evolution of these patients, if there is any data on this.
Reply: The comorbidities reported in the nine included studies varied. No significant difference was found between the remdesivir group and the placebo group.
- Adverse reactions were found?
Reply: The adverse reactions reported in the nine included studies varied, including nausea, diarrhea, hypokalemia, headache, skin rash, liver function dysfunction, renal function impairment, and neutropenia. We conducted this meta-analysis to evaluate mortality among patients treated with remdesivir. Therefore, we did not emphasize the adverse reactions to remdesivir.
Reviewer 2 Report
The manuscript mainly discussed the clinical benefit of hospitalized COVID-19 patients treated with remdesivir, which is related to the ordinal scale at the start of treatment. There are still some points to be addressed.
1. The figure 3 to figure 9 in the manuscript were not simple and clear enough to present the topic and content of this manuscript. If possible, please modify it.
2. The ordinate of figure 2 was not conducive to reading, please modify it.
3. The title and annotation of the figure in line 150 was missing, please modify it.
4. The title of this manuscript was not attractive enough, the author should embody the significance of this finding for COVID-19 treatment.
5. Was the conclusion in the manuscript related with the age of the patient?
The English is fine in this paper.
Author Response
Reviewer 2:
- The figure 3 to figure 9 in the manuscript were not simple and clear enough to present the topic and content of this manuscript. If possible, please modify it.
Reply:
Figure 3 legend: Nine studies including 4381 patients reported hospital mortality or 28-day mortality rates. The mortality rate was significantly different between remdesivir-treated and placebo-treated patients.
Figure 4 legend: Three studies including 656 patients with no supplemental oxygen at the start of treatment reported hospital mortality or 28-day mortality rates. The mortality rate was not significantly different between remdesivir-treated and placebo-treated patients.
Figure 5 legend: Three studies including 1329 patients requiring supplemental low-flow oxygen at the start of treatment reported hospital or 28-day mortality rates. The mortality rate was significantly different between remdesivir-treated and placebo-treated patients.
Figure 6 legend: Three studies including 579 patients requiring supplemental high-flow oxygen or noninvasive mechanical ventilation at the start of treatment reported hospital or 28-day mortality rates. The mortality rate was not significantly different between remdesivir-treated and placebo-treated patients.
Figure 7 legend: Two studies including 397 patients requiring invasive mechanical ventilation or ECMO at the start of treatment reported hospital or 28-day mortality rates. The mortality rate was not significantly different between remdesivir-treated and placebo-treated patients.
Figure 8 legend: Five studies including 2489 patients not requiring supplemental oxygen or requiring supplemental low-flow oxygen at the start of treatment reported hospital or 28-day mortality rates. The mortality rate was significantly different between remdesivir-treated and placebo-treated patients.
Figure 9 legend: Four studies including 1303 patients requiring high-flow supplemental oxygen, non-invasive mechanical ventilation, invasive mechanical ventilation or ECMO at the start of treatment reported hospital or 28-day mortality rates. The mortality rate was not sig-nificantly different between remdesivir-treated and placebo-treated patients.
- The ordinate of figure 2 was not conducive to reading, please modify it.
Reply:
Figure 2 legend: The studies of Mahajan L. and Nevalainen OPO included in the current meta-analysis had a high risk of bias.
- The title and annotation of the figure in line 150 was missing, please modify it.
Reply: Figure 2: Risk of bias of nine randomized controlled trials
- The title of this manuscript was not attractive enough, the author should embody the significance of this finding for COVID-19 treatment.
Reply: I change the title to “No clinical benefit of mortality reduction in hospitalized adult COVID-19 patients who required high-flow supplemental oxygen or invasive mechanical ventilation treated with remdesivir”
- Was the conclusion in the manuscript related with the age of the patient?
Reply: The conclusion in the manuscript was not related to the age of the patient. The mean age of the nine studies included in the current meta-analysis is less than 65 years. The findings of the meta-analysis may primarily pertain to a population of patients who are younger than 65 years old.